# Deep Learning Control for Digital Feedback Systems: Improved Performance with Robustness against Parameter Change

**Nuha A. S. Alwan** [1] and **Zahir M. Hussain** [2,*]

1   College of Engineering, University of Baghdad, Baghdad 10011, Iraq; n.alwan@ieee.org
2   School of Engineering, Edith Cowan University, Joondalup, WA 6027, Australia
*   Correspondence: zmhussain@ieee.org or z.hussain@ecu.edu.au

**Abstract:** Training data for a deep learning (DL) neural network (NN) controller are obtained from the input and output signals of a conventional digital controller that is designed to provide the suitable control signal to a specified plant within a feedback digital control system. It is found that if the DL controller is sufficiently deep (four hidden layers), it can outperform the conventional controller in terms of settling time of the system output transient response to a unit-step reference signal. That is, the DL controller introduces a damping effect. Moreover, it does not need to be retrained to operate with a reference signal of different magnitude, or under system parameter change. Such properties make the DL control more attractive for applications that may undergo parameter variation, such as sensor networks. The promising results of robustness against parameter changes are calling for future research in the direction of robust DL control.

**Keywords:** deep learning; feedback control; conventional controller; neural network; backpropagation; robust control





## 1. Introduction

Design methods for feedback control systems are well-established. These include classical linear control system design, techniques for nonlinear control, robust control, H-∞ control and adaptive control. In addition, model-free control has emerged with techniques such as fuzzy logic control and artificial neural networks (ANN). The latter methods generally extend adaptive control techniques to nonlinear systems [1]. Closed-loop control applications of NNs are different from classification and image processing applications, which are open-loop. NNs were first introduced in closed-loop control systems by Werbos in [2]. Offline learning, in particular, was formalized in [3] and was shown to yield important structural information. In addition, an important problem that also had to be addressed in closed-loop NN control was weight initialization for feedback stability [4]. In this work, an NN is trained and used for function approximation to replace a conventional controller in a digital feedback control system. Neural networks can model linear or nonlinear systems as they are excellent at finding the underlying processes that govern these systems. It has been stated in [1] that the two-layer NN is sufficient for feedback control purposes. In the present work, however, we show that the addition of hidden layers, resulting in deep NN controllers, produces a damping effect and improves feedback control system stability. The multi-layer NN (specifically the two-layer network with one hidden layer) took 30 years to effectively replace the single-layer NN, which was introduced as early as the 1950s. The reason was the lack of the proper learning rule to update the hidden layer weights during training, a problem that was later solved by the backpropagation (BP) algorithm in 1986 [5]. The BP algorithm is also based on stochastic gradient descent (SGD) learning as in the single-layer NN, but uses a different method to update the gradient, namely, backpropagation. It took another 20 years to solve the poor

performance issues of the deep NN (two or more hidden layers) through the innovation of deep learning (DL). DL, in essence, comprises many small technical improvements to ensure proper training. DL solves problems such as vanishing gradient, overfitting and computational load [6,7]. A deep NN with two hidden layers is shown in Figure 1. The inter-layer arrows in the figure represent weighted connections. It is a multiple-input, multiple-output (MIMO) system; however, in this work, only a single outlet is considered, with the hidden layers using a nonlinear activation function $\Phi(\cdot)$, while the output layer uses a linear activation function $\Psi(\cdot)$. The output vector $Y$ in Figure 1 can be expressed as follows:

$$Y_{L \times 1} = \Psi \left[ M_{L \times K} \, \Phi \left\{ H_{K \times J} \, \Phi \left( W_{J \times I} \, X_{I \times 1} + B_{J \times 1} \right) + \beta_{K \times 1} \right\} + b_{L \times 1} \right] \tag{1}$$

where $X$ is the input vector; $W$, $H$ and $M$ are the weight matrices of the first hidden, second hidden and output layers, respectively; $I$, $J$, $K$ and $L$ are the numbers of nodes of the input, first hidden, second hidden and output layers, respectively. The variables $B$, $\beta$ and $b$ represent the respective biases. The activation functions operate point-wise on the relevant vectors.

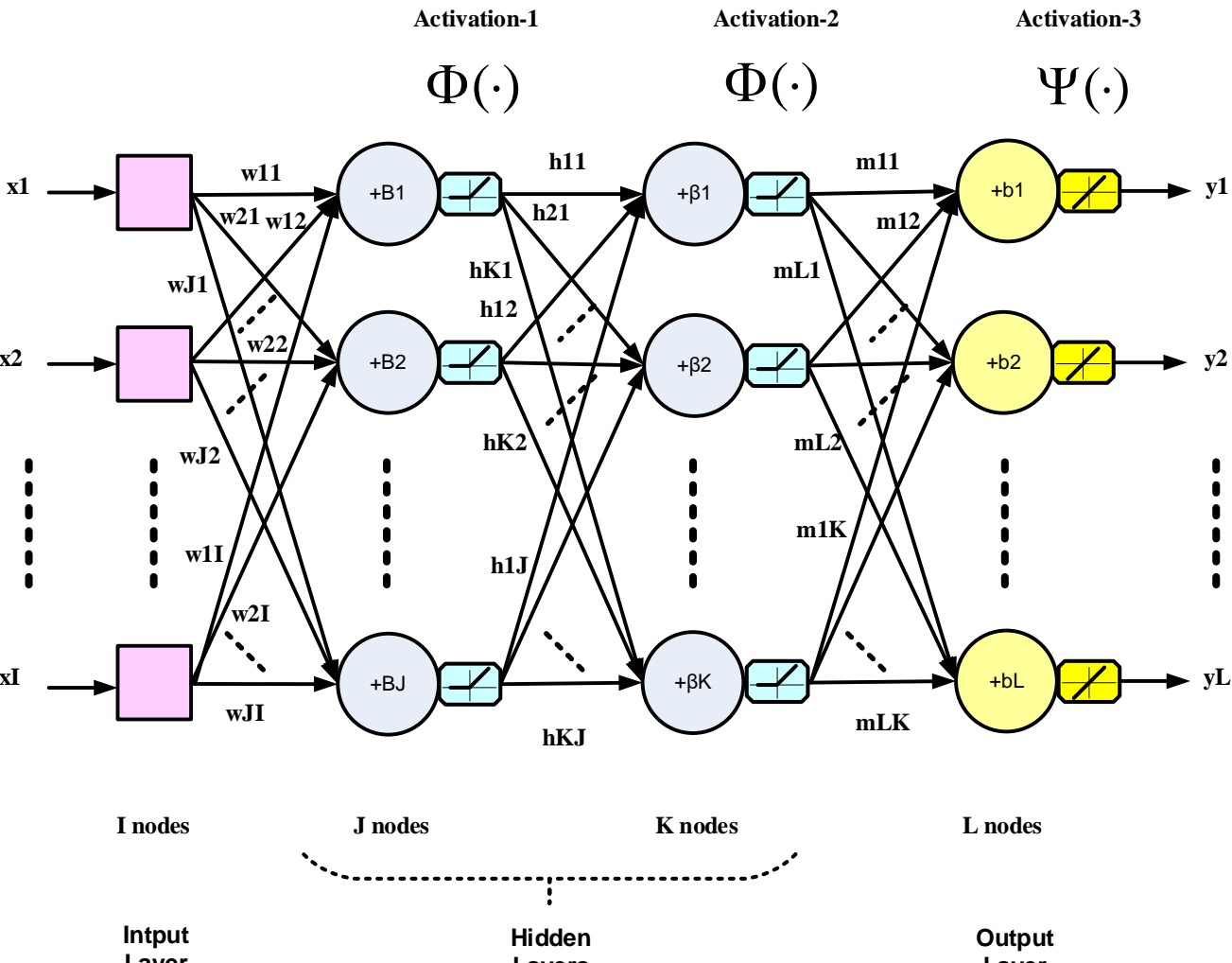

**Figure 1.** A generic diagram for a three-layer deep NN (neural network) with two hidden layers.

In this work, a conventional controller is first designed and tested in a feedback control system. By using the input-output information of this controller as the training data of a learning algorithm such as the BP algorithm, a DL controller consisting of a deep NN is trained offline and then made to replace the conventional controller. Finally, the system or

plant is controlled just by the DL controller and its performance monitored. It is shown that the feedback control system performance criteria can all be improved such as settling time, overshoot, steady state error, etc.

Relevant recent works in the literature include [8] where a NN controller with one hidden layer is trained for use with multi-input, multi-output (MIMO) systems resulting in an improvement in transient response regarding overshoot and settling time, but without resorting to DL. In [9], the speed of a DC motor is controlled in a feedback control loop with a proportional-integral-derivative (PID) controller, the most commonly used in industry. Similar to the present work, DL is also resorted to in order to design an intelligent controller but via a deep belief network (DBN) algorithm [10]. A DBN performs a kind of unsupervised learning using a restricted Boltzmann machine (RBM) [10] to generate a set of initial weights to improve learning. RBM's are networks in which probabilistic states are learned for a set of inputs suitable for unsupervised learning. A similar approach to DL control is the work in [11], where RBMs are also used for weight initialization by unsupervised training. The disadvantage of DBNs is the hardware requirement since they consist of two stages, unsupervised pre-training and supervised fine tuning. The ordinary deep NNs used in the present work are less computationally demanding, and moreover, single-stage supervised offline learning is possible due to the availability of input-target data in the application considered.

In addition to supervised and unsupervised learning, there is also a third type of learning in machine learning called reinforcement learning (RL). This is learning by making and correcting mistakes in a trial-and-error fashion, that is, learning by experience in case of the absence of a training data set. It is a process in which a software agent makes observations and takes actions within an environment and in return, it receives rewards [12]. RL thereby achieves long-term results, which are otherwise very difficult to achieve. Deep RL has recently been used in robotic manipulation controllers [13,14]. A deep learning controller based on RL is also implemented in [15] for the application of DL in industrial process control. However, RL is very computationally expensive and requires large amounts of data, and as such, it is not preferable to use to solve simple problems. RL suffers from the lack of real-world samples such as in robotic control where robotic hardware is expensive and undergoes wear and tear.

In [16], a deep NN controller is developed to reduce the computational cost of implementing model predictive control (MPC). MPC is a reliable control strategy to effect control actions by solving an optimization problem in real time. However, the deep NN controller architecture uses long short-term memory-supported NN (LSTMSNN) models, and therefore needs a graphical processing unit (GPU) to accelerate the online implementation of the controller by parallelizing computations. Similarly, ref. [17] uses DL-based techniques and recurrent NNs for real-time embedded implementation of MPC, thereby eliminating online optimization.

In the present work, the deep NN controller used with a second-order plant in a closed-loop control system is more computationally efficient in the training phase as well as in real-time operation, while retaining DL benefits, compared to the above works that employ DBNs, RL and LSTMSNNs. In addition, detailed results are presented regarding improvement in settling times as the DL controller implementation gets deeper, steady state error and overshoot. The results of using different activation functions in hidden layers have also been considered. Effects of parameter changes such as changes in plant gain and pole locations are elaborated on to prove the robustness of the present approach versus conventional electronic design. Robust design of control systems is necessary to keep the plant performance optimal under parameter variation [18]. However, robustness of the proposed DL has only been tested against its electronic counterpart, but has not been tested for optimality in the sense of optimal robust control. The authors hope that future research in this direction can succeed in extending the optimization methods in [18] to develop an optimized method for DL robust control. In short, the present work offers a versatile experimental setup of a standard deep NN controller in digital feedback control

systems, thus unraveling several aspects regarding the vast potential of machine learning and especially DL techniques in process control.

The rest of the paper is organized as follows: Section 2 presents the design of the conventional controller within the digital feedback control system. Section 3 explains the training procedure of the deep NN to construct the DL controller. The simulation results are given in Section 4, and finally, Section 5 concludes the paper.

## 2. Conventional Controller Design for Digital Feedback Control System

The block diagram of a feedback control system with a digital controller $D(z)$ is shown in Figure 2. The z-transforms of the input sampled reference signal, the sampled error signal, the sampled control signal and the sampled output signal are denoted by $R(z)$, $E(z)$, $U(z)$ and $Y(z)$, respectively. We assume that $G(z)$ is the z-transform of $G(s)$ which is given by:

$$G(s) = G_o(s) \cdot G_p(s) \tag{2}$$

where $G_p(s)$ is the plant transfer function and $G_o(s)$ is the transfer function of the zero-order hold, which represents a digital-to-analog converter that converts the sampled control signal to a continuous signal to be input to the plant. The zero-order hold takes the sample value and holds it constant for the duration of the sampling interval $T$. As such, $G_o(s)$ is given by [19]:

$$G_o(s) = \frac{1}{s} - \frac{1}{s} e^{-sT} \tag{3}$$

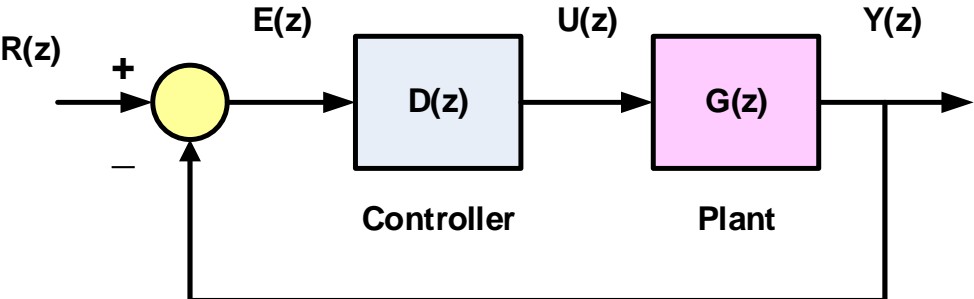

**Figure 2.** Block diagram of a feedback control system with a digital controller.

The closed-loop transfer function is:

$$\frac{Y(z)}{R(z)} = T(z) = \frac{G(z)\,D(z)}{1 + G(z)\,D(z)} \tag{4}$$

We consider a second-order system with the plant transfer function given by:

$$G_p(s) = \frac{6960}{s(s+4)} \tag{5}$$

Note that, whereas a continuous second-order feedback control system is stable for all values of gain assuming left-half s-plane open-loop poles, a second-order sampled system can be unstable with increasing gain [19]. Using Equations (2), (3) and (5), we convert to digital as follows:

$$\begin{aligned}
G(z) &= Z[G(s)] = Z\left[\frac{1-e^{-sT}}{s}\,\frac{6960}{s(s+4)}\right] \\
&= (1-z^{-1})\,Z\left[\frac{6960}{s^2\,(s+4)}\right] \\
&= 6960\,(1-z^{-1})\,Z\left[\frac{1}{4s^2} - \frac{1}{16\,s} + \frac{1}{16\,(s+4)}\right] \\
&= 1740\,(1-z^{-1})\left[T\frac{z}{(z-1)^2} - \frac{1}{4}\,\frac{z}{(z-1)} + \frac{1}{4}\,\frac{z}{(z-e^{-4T})}\right]
\end{aligned} \tag{6}$$

Substituting in the above for $T$ by 0.001 s, we obtain:

$$G(z) = \frac{0.003475\, z + 0.003471}{z^2 - 1.996\, z + 0.996} \tag{7}$$

An analog controller $G_c(s)$ can be designed as a lead compensator such as to achieve a phase margin of 45° with a crossover frequency of 125 rad/s. Using the compensation design methods in [19] for meeting phase margin specifications, we find that:

$$G_c(s) = \frac{5.6\,(s + 50)}{s + 312} \tag{8}$$

With $T = 0.001$ s, we find $D(z) = Z[G_c(s)]$ as:

$$D(z) = \frac{4.85\, z - 4.61}{z - 0.73} \tag{9}$$

Now, the system in Figure 2 can be implemented in MATLAB with a unit-step reference input in order to obtain the digital error and control signals, $e(n)$ and $u(n)$, respectively. These signals are shown in Figures 3 and 4, whereas the unit step response of the control system is shown in Figure 5. While these figures are simulation results, an analytical expression for the unit step response can be obtained. In Equation (4), substituting for $G(z)$ and $D(z)$ by Equations (7) and (9), respectively, and also substituting for the unit step reference input $R(z)$ by $z/(z-1)$, we solve by partial fractions to obtain:

$$Y(z) = \frac{z}{z - 1} + \frac{1.9765\, z}{z - 0.8979} + \frac{(-1.4798 + j\,0.181)\, z}{z - (0.9056 + j\,0.0862)} + \frac{(-1.4798 - j\,0.181)\, z}{z - (0.9056 - j\,0.0862)} \tag{10}$$

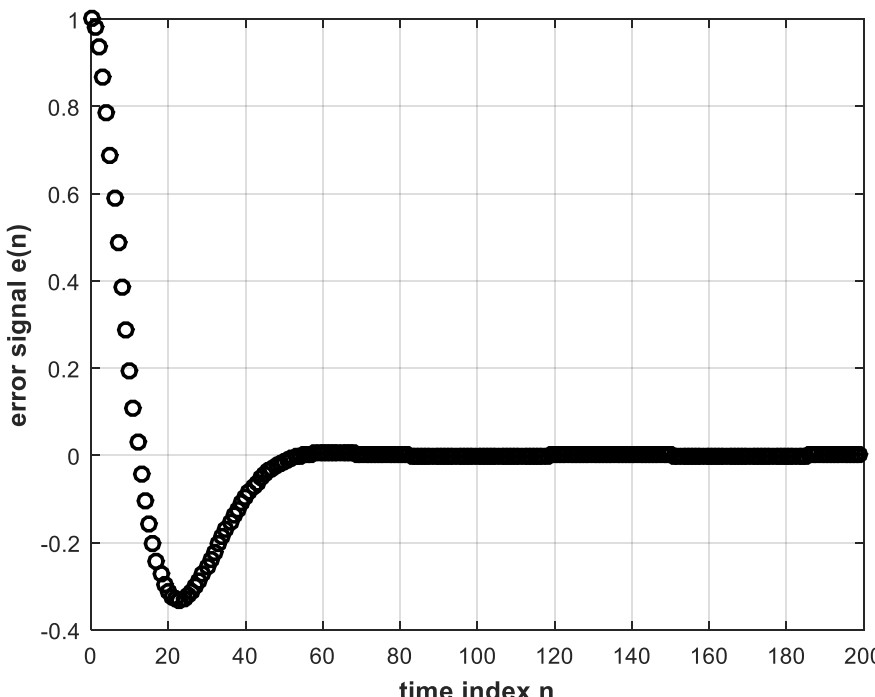

**Figure 3.** The error signal input to the digital controller.

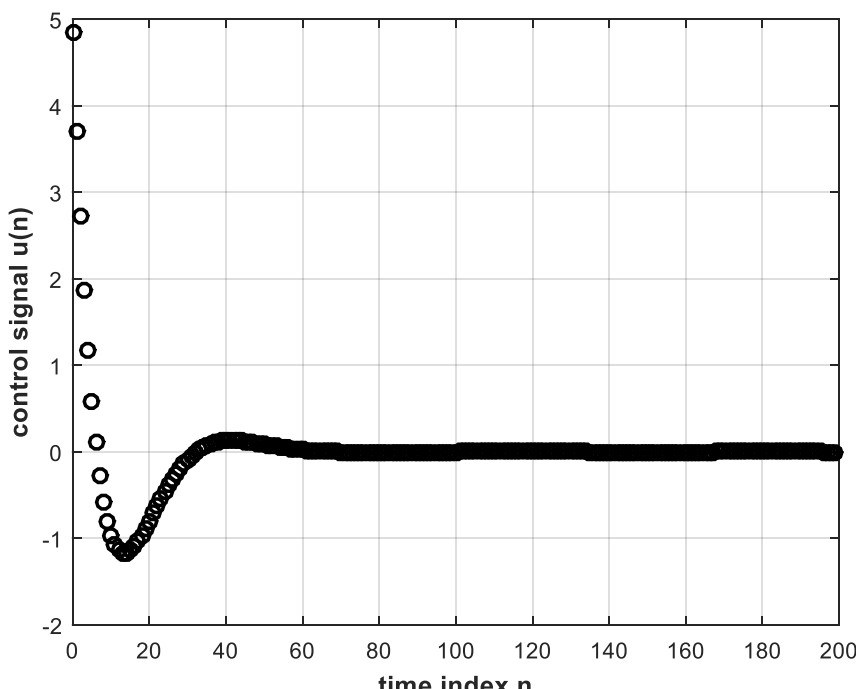

**Figure 4.** The control signal output from the digital controller.

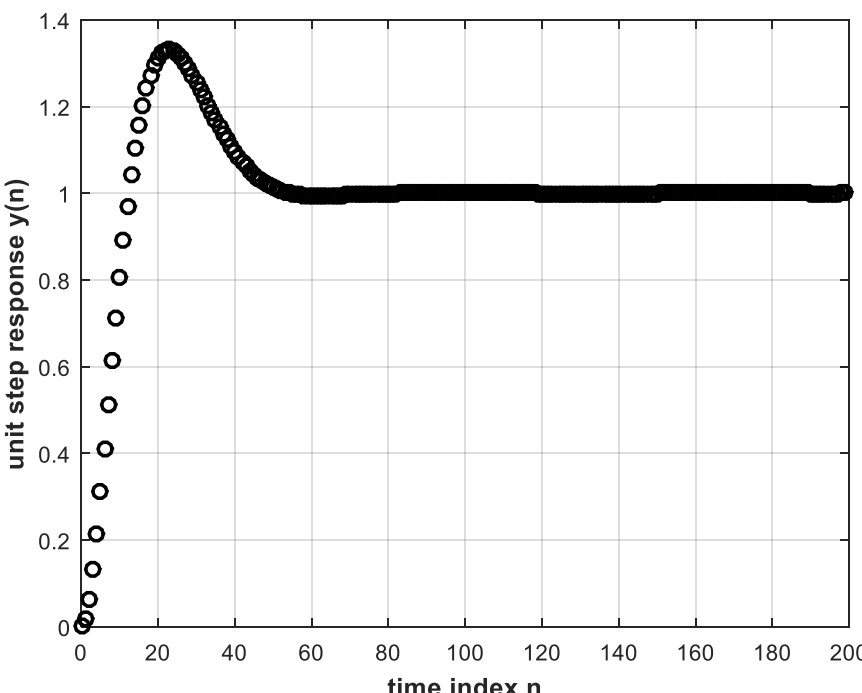

**Figure 5.** The unit step response of the feedback control system.

Taking the inverse Z-transform, we obtain the unit step response as:

$$y(n) = 1 + 1.9765(0.8979)^n + 2.9816(0.9096)^n \cos(0.0302\,\pi\,n + 0.9615\,\pi)\,;\, n \geq 0 \quad (11)$$

Plotting the above expression coincides exactly with the plot of Figure 5. The conventional controller input-output signals are needed to train the DL controller described in the following section.

## 3. The DL Controller

The error and control signals of Figures 3 and 4 can be used to train a deep NN offline to obtain a DL version of the controller that learns the correspondence between $e(n)$ and $u(n)$. In other words, the training data consisting of input and correct output of the DL controller are the error input and control output signals of the conventional controller stored for a sufficient number of discrete time instants during feedback digital control system operation. Training is performed using the BP algorithm [6]. The trained DL controller is intended to replace the conventional controller for real-time feedback control system operation.

There are several training algorithms for training the neural network weights, the most important being the backpropagation (BP) algorithm, in which the output error starts from the output layer and propagates backwards until it reaches the hidden layer next to the input layer to update the weights. Based on the update strategy, there are different variations of BP, the most common being BP based on gradient descent (GD) [20]. With reference to Figure 1, the GD-based BP algorithm for updating the weights can be summarized by the following equations, assuming a linear activation function for the output layer and nonlinear activation functions for the hidden layers.

$$
\begin{aligned}
&m_{lk} \leftarrow m_{lk} + \Delta m_{lk}\,, l = 1, \ldots, L \text{ and } k = 1, \ldots, K.\\
&\text{where } \Delta m_{lk} = \alpha\,(d_l - y_l)\,y_k = \alpha\,\delta_l\,y_k\\
&\quad h_{kj} \leftarrow h_{kj} + \Delta h_{kj}\,, k = 1, \ldots., K \text{ and } j = 1, \ldots., J.\\
&\text{where } \Delta h_{kj} = \alpha\,\delta_k\,y_j \text{ with } \delta_k = \left[\sum_l m_{lk}\,\delta_l\right]\cdot\Phi'(v_k)\\
&\quad w_{ji} \leftarrow w_{ji} + \Delta w_{ji}\,, j = 1, \ldots., J \text{ and } i = 1, \ldots., I.\\
&\text{where } \Delta w_{ji} = \alpha\,\delta_j\,x_i \text{ with } \delta_j = \left[\sum_k h_{kj}\,\delta_k\right]\cdot\Phi'(v_j)
\end{aligned}
\tag{12}
$$

and so on for more hidden layers. In Equation (12) above, $\alpha$ is the BP learning rate, the $d's$ are the correct outputs at the output layer needed for supervised training, the $v's$ are the activation function inputs, the $y's$ are the activation function outputs for output and hidden layers, the $x's$ are the inputs to the input layer and $\Phi'(.)$ is the derivative of the nonlinear activation function $\Phi(.)$.

In the present application, the training data consisting of the correct outputs ($d's$) and inputs ($x's$) are values of the control signal $u(n)$ and the error signal $e(n)$, respectively.

It should be noted, however, that the correspondence between $e(n)$ and $u(n)$ is not one-to-one, as can be seen from Figures 3 and 4. Instances can be found of a single value of $e(n)$, occurring at different time indices, that corresponds to more than one value of $u(n)$. These identical values of $e(n)$ at different time indices can be made distinguishable from each other if their past values are taken into account. Error values may be the same for different time instants, but their past behaviors are normally different. Therefore, the input layer of the deep NN may consist of more than one node to take into consideration present as well as past values of the error signal, but the output layer has only one node such that the NN performs regression or function approximation. The activation functions of the hidden-layer nodes are chosen as the rectified linear unit (ReLU) function, which is known to perform better with DL than the sigmoid function [6,7]. The reason is that the ReLU function solves the problem of vanishing gradient due to back-propagating the error during training. The output layer single node, however, can be chosen to have a linear activation function. This would avoid any restriction in amplitude of the controller output which constitutes the control signal to the plant. The sigmoid and tanh functions would restrict the control signal from 0 to 1 and from $-1$ to 1, respectively. Similarly, a ReLU at the output would not allow negative values, whereas the control signal can take on negative as well as positive values as can be discerned from Figure 4. A neural network with two or more hidden layers is considered deep [7]. It is important to properly choose the convenient topology of a NN. Clearly, the number of neurons in the input layer is equal to the number of inputs, and the number of neurons in the output layer is likewise equal to the number of

outputs. As for the number of neurons in the hidden layers, this is a problem facing many researchers. Rules of thumb are given in many instances in the literature [21,22]. Some of these rules state that the number of nodes in the hidden layer should lie between those in the input and output layers, and the number of hidden nodes could be taken as two-thirds the sum of input and output layer nodes. In [23], it is concluded that the power of the NN does not depend on whether the first hidden layer or second hidden layer has more neurons, whereas [24] presents an empirical study of a three-layer (two hidden layers) NN concluding that:

$$n_1 = \lceil 0.5\,n_h + 1 \rceil \quad and \quad n_2 = n_h - n_1 \tag{13}$$

where $n_1$ and $n_2$ are numbers of neurons in the first hidden and second hidden layers, respectively, $\lceil \cdot \rceil$ is the ceil integer function and $n_h$ is the total number of hidden neurons in both hidden layers. The trial-and-error approach, however, is often used to successfully determine the number of layers and the number of neurons in the hidden layers [23].

## 4. Simulation Results and Discussion

All simulations are implemented in MATLAB (Academic License no. 904939). The second-order system with the plant transfer function given by Equation (5) is considered in this study, with analog controller given by Equation (8). The digital counterparts of this plant and its control sub-system (as shown in Figure 2) have been implemented using Equations (7) and (9) in the first round of the simulation, while the second round used control data from this system to train a DL network that eventually replaced the electronic control sub-system of Equation (9). This section will investigate the performance of the DL controller versus the original electronic controller.

### 4.1. DL Training Process

Figures 3 and 4 demonstrate the training data that are used in offline training of the deep NN that constitutes our DL controller. The training data are the necessary input-correct output pairs needed for implementing the BP algorithm which is a supervised learning algorithm. The learning rate of the BP algorithm is taken by trial and error as 0.02. In each training or learning iteration, seven past samples plus the present sample of the error signal are used as input to the deep NN as discussed in Section 3. Thus, the input layer will consist of eight nodes. By trial and error, this number of necessary input nodes was found to yield optimum performance for this case of two hidden layers. The output layer will have only one node as only one controller output is required to provide the control signal to the plant. The output node is made to operate with a linear activation function. The number of neurons in each of the hidden layers is taken as five. For two hidden layers, this number is almost in conformity with the empirical rule of Equation (13). The nonlinear activation function for all hidden nodes is the ReLU function which has the merit of solving the vanishing gradient problem for deep NNs trained by the BP algorithm [6,7]. As in Figures 3 and 4, the number of available training data pairs is 200. Training for 200 iterations, called an epoch [7], is repeated 200 times. That is, the number of epochs is also 200. The DL controller with two hidden layers is first trained. Its weights are initialized with real random numbers between 1 and −1 taken from a uniform random distribution. However, the learning behavior and subsequent closed-loop stability was found to be sensitive to the initial values of the weights. Therefore, different initial random weights from the same uniform distribution were tried for good performance. The offline nature of our training procedure renders this possible. There exist, however, various weight initialization methods and algorithms in the literature [25,26]. These can prove especially helpful in applications where the controller parameters are to be adjusted online [27]. When learning was achieved, the DL controller was used in inference mode to test its performance. In this mode, the same stored error signal from the conventional controller is entered to the trained DL controller, every sample with its seven past values, and the DL controller output is obtained as shown in Figure 6. It is clear that good learning is achieved as Figure 6 is almost identical to Figure 4.

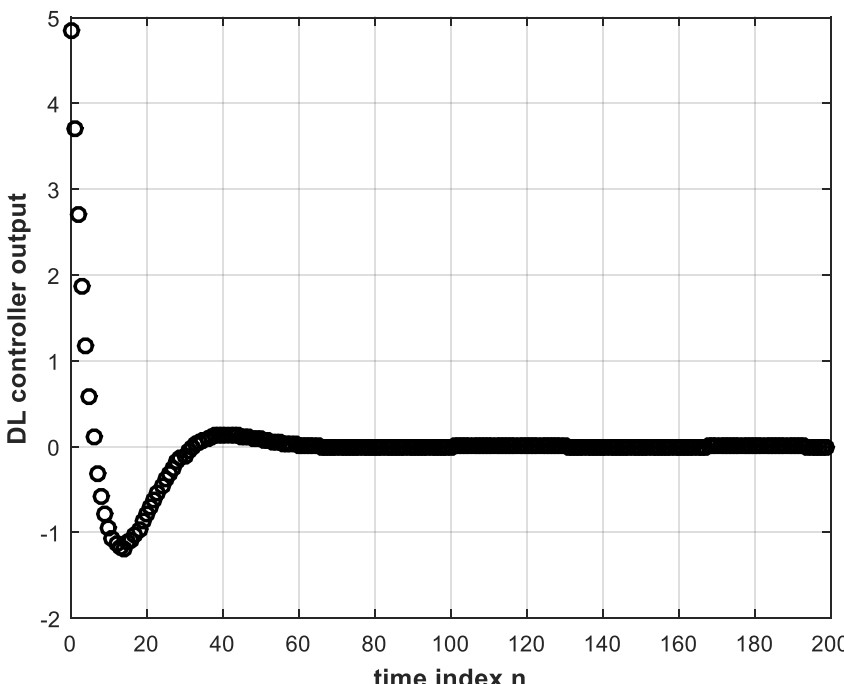

**Figure 6.** Output signal of trained DL controller (with two hidden layers) in inference mode.

### 4.2. Performance of the DL Controller versus Electronic Controller

The next step is to use the trained DL controller in real time to provide the control signal to the plant within the feedback control system. The unit step response is shown in Figure 7. This, in turn, is almost identical to Figure 5 that corresponds to the conventional controller whose behavior was learned.

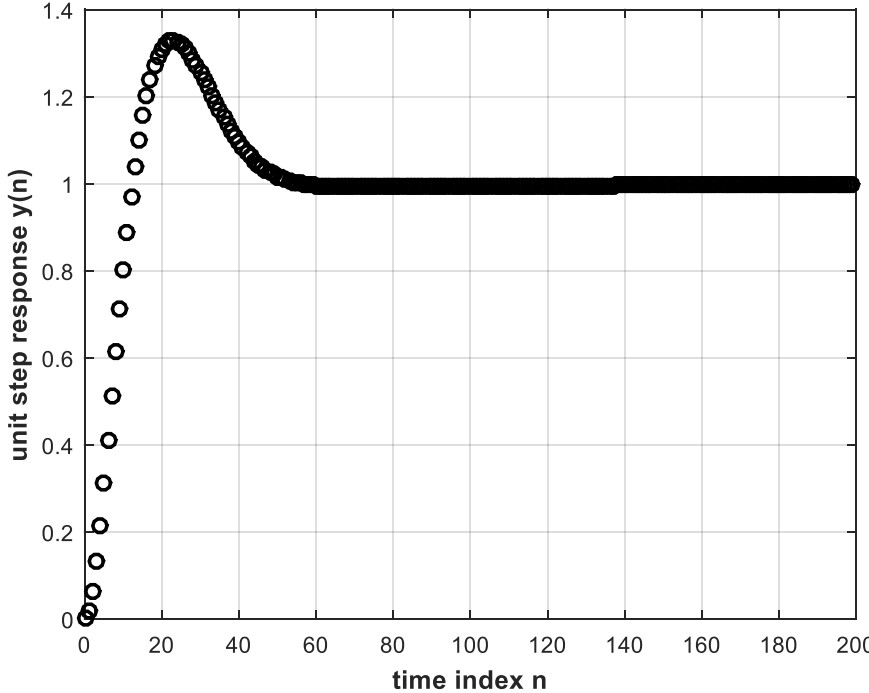

**Figure 7.** Unit step response of feedback control system using the trained DL controller with two hidden layers.

The same trained DL controller is made to operate within the feedback system but subject to a step reference of magnitude 2. The resulting step response of the system is

shown in Figure 8, where it is clear that the system yields the correct response with the same trained DL controller regardless of the step magnitude. Note that the input to the DL controller is different from the training input. That is, the DL controller does not exhibit any signs of overfitting [6]. The latter is a situation where the deep NN fails to respond correctly to other than the training data. That is why, in this work, we do not need to employ dropout which is another DL technique used to overcome overfitting [6,7].

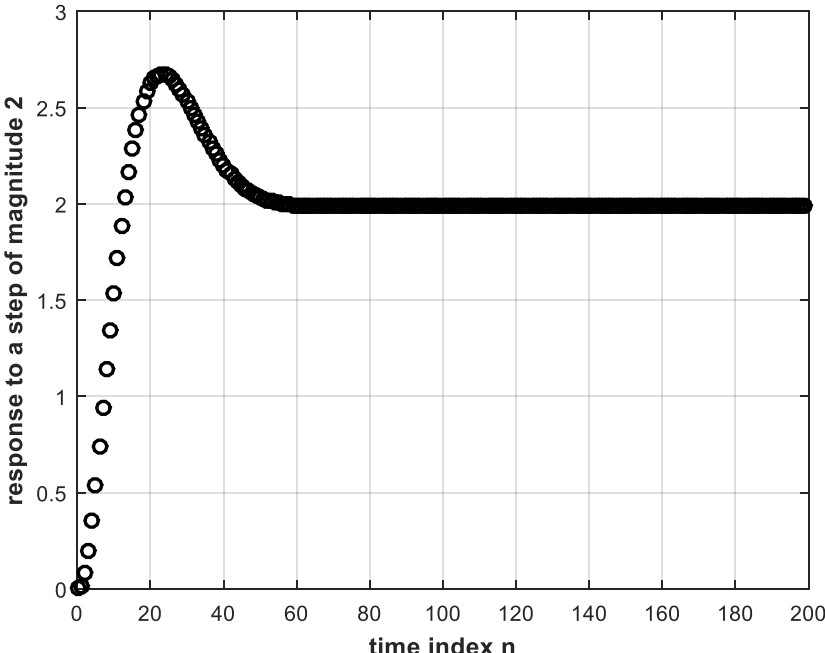

**Figure 8.** System response to a step magnitude of 2 using the trained DL controller.

We now investigate the use of deeper NNs for DL controllers in feedback control and compare between them as well as with the conventional controller in terms of settling time. The settling time of the second-order system under consideration is taken as the time needed for the system step response to reach a value within a certain percentage of the final value. The percentages considered are 2, 0.2 and 0.02. Table 1 lists the settling time of the different tested cases. The number of nodes is five in each hidden layer, and the number of input layer nodes is eight for all cases. It can be seen from Table 1 that the deeper the controller the smaller (better) the settling time of the feedback control system. The reason is that better learning is achieved with deeper NNs. Moreover, with four hidden layers, the DL controller even outperforms the conventional controller in terms of settling time. Thus, the DL controller has the advantage of producing a damping effect when sufficiently deep, thereby enhancing performance.

**Table 1.** Unit step response settling times for the different types of controllers under consideration.

| Controller Type | Settling Times (Milliseconds) within r % of Final Value | | |
|---|---|---|---|
| | r = 2 | r = 0.2 | r = 0.02 |
| Conventional | 48 | 74 | 104 |
| DL with 2 hidden layers | 49 | >200 | >200 |
| DL with 3 hidden layers | 49 | 85 | 120 |
| DL with 4 hidden layers | 47 | 55 | 92 |

The number of input layer nodes was fixed at eight for all entries of Table 1 for the purpose of comparison. However, with four hidden layers, results are better if we set this number to four. For this case, the feedback system unit step responses with the conventional and trained DL controllers are shown in Figure 9.

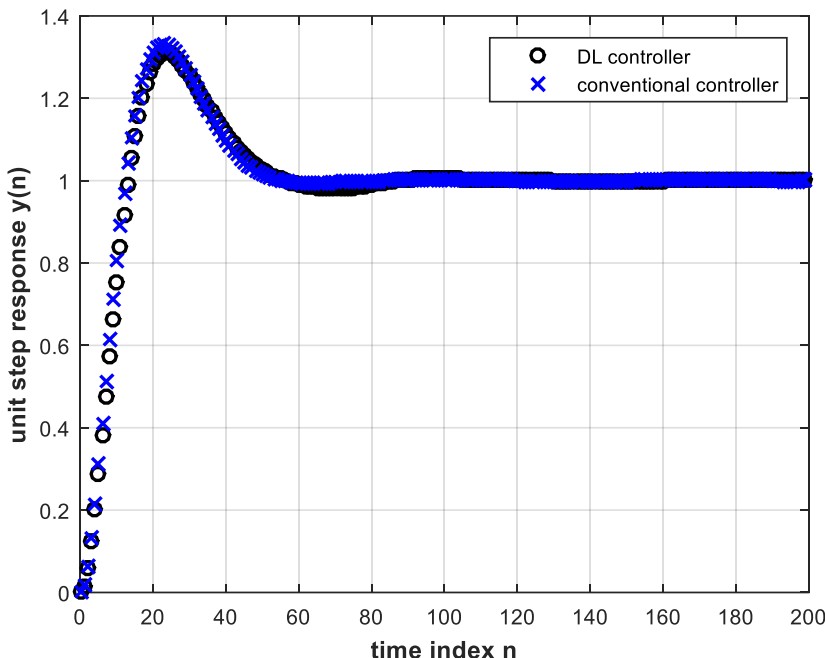

**Figure 9.** Unit step response of the feedback system with conventional and trained DL controllers. The number of hidden layers is four and the number of input nodes is four.

*4.3. Robustness against System Parameter Change*

Next, we investigate the influence of system parameter change on the performance of the trained DL controller versus its conventional electronic counterpart. In all plants, process parameters such as the properties of materials including thermal and electrical conductivity, dimensions of components such as distances between capacitive plates, etc. may change with time due to aging, pressure, vibration, corrosion and so on. A control system is robust when these changes have little or no effect on its performance. Otherwise, these changes worsen the system performance. Changes of plant gain and pole locations will be considered separately as follows.

4.3.1. Performance under Plant Gain Change

Let us assume that the plant gain in Equation (5) undergoes a considerable change, from 6960 to 20,000. Using Equation (2) and converting to digital with $T = 0.001$ s, we arrive at the following expression for $G(z)$ using the same steps that led to Equation (7):

$$G(z) = \frac{0.009987\,z + 0.009973}{z^2 - 1.996\,z + 0.996} \tag{14}$$

The trained DL controller for Figure 9 with four hidden layers and four input nodes is tested in online operation of the feedback control system with the above plant parameter change. The unit-step response is shown in Figure 10 together with the simulated conventional controller case (without re-design) under the same system parameter change.

Proceeding with a workout similar to that which led to Equation (11), the analytical expression for the feedback system unit step response using the conventional controller of Equation (9) under system parameter change is:

$$y(n) = 0.9999 + 0.1433(0.9439)^n + 1.0948(0.9050)^n \cos(0.0927\,\pi\,n + 0.9936\,\pi); \ n \geq 0 \tag{15}$$

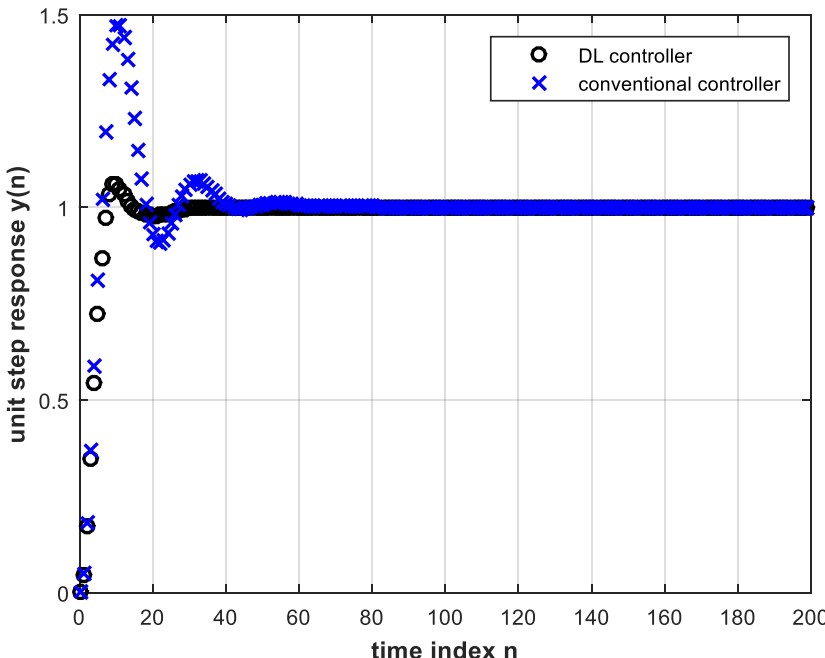

**Figure 10.** Unit step response of the feedback control system with the trained DL controller (four hidden layers and four input nodes) as well as with the conventional controller, both under plant gain change, Equation (14).

Plotting Equation (15) above coincides exactly with the corresponding plot in Figure 10. It is clear from Figure 10 that the DL controller behaves much more satisfactorily than the conventional controller. This means that the latter has to be redesigned for successful operation, whereas the DL controller does not need re-training. A magnified view of Figure 10 is shown in Figure 11.

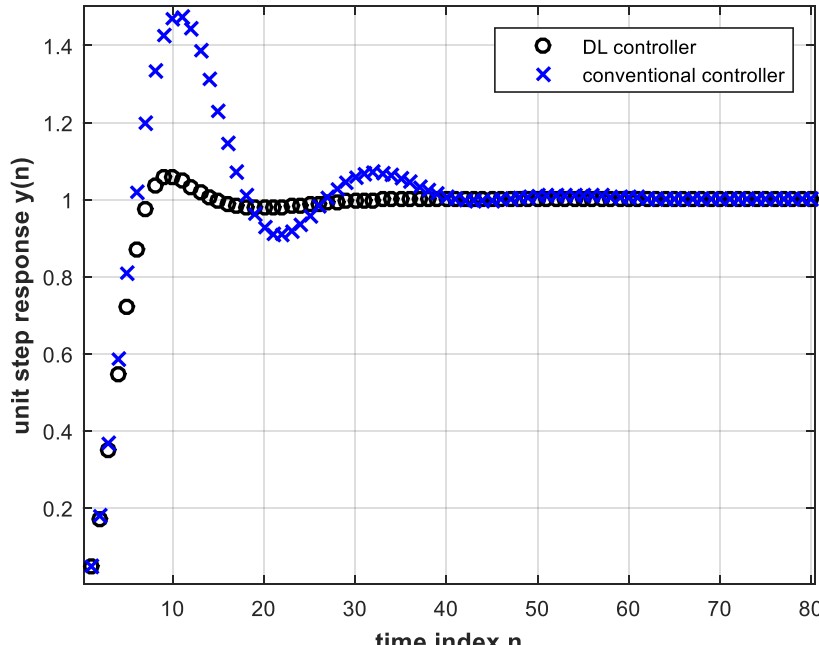

**Figure 11.** A magnified view of Figure 10.

The advantage of obtaining smaller settling time with the DL controller is evident. DL causes the system to reach steady state faster with less overshoot; the first-peak ratio is almost 8, where the first-peak ratio $\Re$ is defined as the ratio of the first transient response

peak using the conventional controller to that using the DL controller. Table 2 shows the settling times of the control system for both controllers. The steady state error for all above results is zero.

**Table 2.** Unit step response settling times for different controllers under plant gain change.

| Controller Type | Settling Times (Milliseconds) within r % of Final Value | | |
|---|---|---|---|
| | r = 2 | r = 0.2 | r = 0.02 |
| Conventional | 40 | 80 | 114 |
| DL with 4 hidden layers | 14 | 30 | 43 |

### 4.3.2. Performance under Pole Location Change

If the pole locations of the analog plant change further towards instability, the closed-loop digital control system unit step response begins to exhibit steady state error while using the same trained DL controller and the same conventional controller. This is shown in Figure 12 and the magnified view of Figure 13 for the analog plant given by:

$$G_p(s) = \frac{6960}{s(s+0.5)} \tag{16}$$

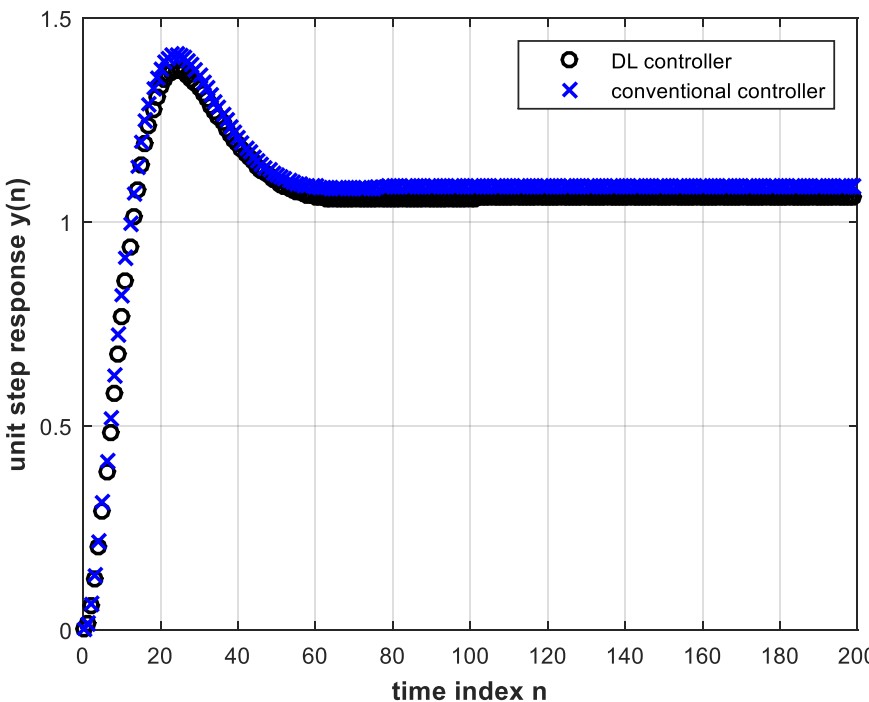

**Figure 12.** Unit step response of the feedback control system with the trained DL controller (four hidden layers and four input nodes) as well as with the conventional controller, both under pole location change, Equation (17).

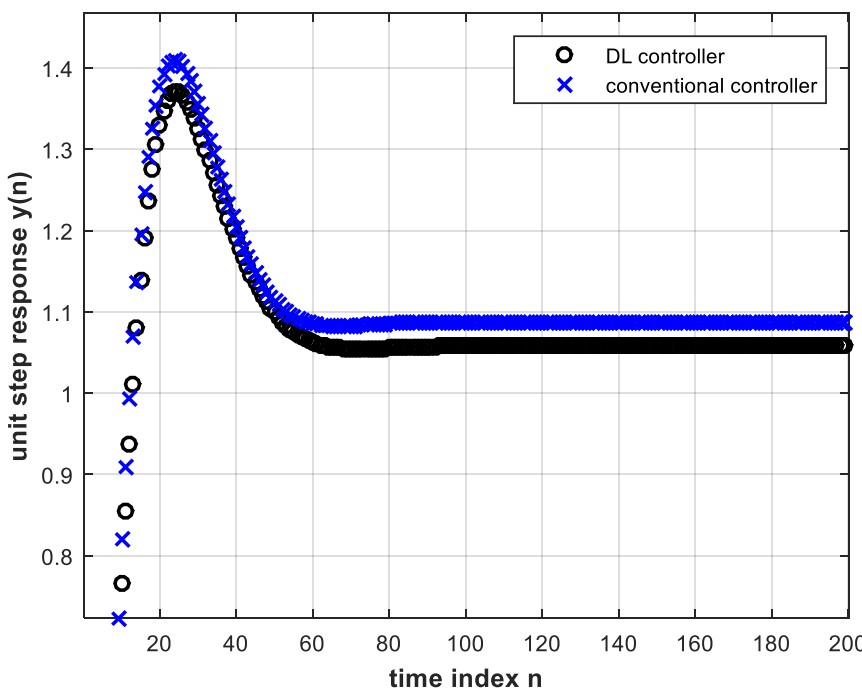

**Figure 13.** A magnified view of Figure 12.

This, together with the zero-order hold, and with $T = 0.001$ s corresponds to the digital system given by:

$$G(z) = \frac{0.003479\,z + 0.003479}{z^2 - 2\,z + 0.9995} \tag{17}$$

Under this parameter change, the analytical expression for the unit step response is:

$$y(n) = 1.0901 + 2.1642(0.9011)^n + 3.2810(0.9096)^n \cos(0.0285\,\pi\,n + 0.9480\,\pi); \; n \geq 0 \tag{18}$$

Figure 13 shows a steady state error for the conventional controller that is greater than that for the DL controller which is another advantage of the latter. The DL controller also results in smaller overshoot. By the final value theorem, the steady state error of the digital feedback control system subject to a unit step input, and using the conventional controller, is given by:

$$e_{ss} = \lim_{z \to 1} \left\{ (1 - z^{-1}) \frac{1}{1 + D(z)\,G(z)}\,R(z) \right\} \tag{19}$$

where

$$R(z) = \frac{1}{1 - z^{-1}} \tag{20}$$

Therefore,

$$e_{ss} = \lim_{z \to 1} \left\{ \frac{1}{1 + D(z)\,G(z)} \right\} \tag{21}$$

Substituting in the above by Equations (9) and (17), we find that $e_{ss} = -0.09$. This is in accordance with the curve in Figure 13 that corresponds to the conventional controller and with Equation (18) as n tends to infinity.

The relative percentage change in pole value is defined as:

$$\rho = \frac{\Delta p}{p_i} \times 100\% = \frac{p_f - p_i}{p_i} \times 100\% \tag{22}$$

where $p_f$ is the final pole value and $p_i$ is the initial pole value. For the above pole change from $s = -4$ to $s = -0.5$, and using Equation (22), $\rho$ would be $-87.5\%$. Table 3 below lists the steady state error values versus $\rho$ for both controllers as $\rho$ changes from $-75\%$ to nearly

−100%. The other pole location is fixed at $s = 0$. It is clear that the steady state error is always better with the DL controller.

**Table 3.** Steady state error versus percentage pole change for different controllers.

| Relative Percentage Pole Change ($\rho$) | Steady State Error | |
|---|---|---|
| | **(DL Controller)** | **(Conventional Controller)** |
| −75 | 0 | 0 |
| −77.5 | +0.01 | +0.02 |
| −87.5 | −0.05 | −0.09 |
| −90 | −0.045 | −0.07 |
| −92.5 | −0.03 | −0.05 |
| −95 | −0.02 | −0.035 |
| −97.5 | −0.011 | −0.017 |

Positive values of $\rho$ result when the pole moves further away from the imaginary axis in the s-plane. In this case, both analog systems and digital counterparts perform well with zero steady state error so that the benefit of the DL controller is not noticeable. Therefore, positive values of $\rho$ will not be considered further.

A double-pole change is also considered by changing $s = −4$ to $s = −2$ and $s = 0$ to $s = −0.5$. In this case, the DL controller also exhibited improved performance regarding both steady state error and overshoot, as compared to the conventional controller.

*4.4. Effect of Activation Functions*

It is useful here to consider the possibility of using different activation functions in different hidden layers and present a corresponding numerical comparison related to the plant gain change experiment in Section 4.3.1. It is found that the DL controller with four hidden layers fails to train if sigmoid activation functions are used in all layers due to the vanishing gradient problem [6]. When only the first two hidden layers were assigned sigmoid activation functions and the remaining layers had ReLU functions, the DL controller also failed to train properly for the same previous reason. However, when using ReLU functions in the first two hidden layers and sigmoid functions in the outer two hidden layers, training was possible and DL benefits were retained, but not as markedly as in the case of all-ReLU functions. The training is still possible because the first two hidden layers with ReLU functions save the gradient reaching them from vanishing to zero during backpropagation. For a numerical comparison relating to the above argument, we use the first-peak ratio as a performance measure. It is found that the first-peak ratio $\Re$ that was 8 under plant gain change is now reduced to only $\Re = 2$ under the same change when the DL controller hidden layers use the activation function arrangement of ReLU-ReLU-sigmoid-sigmoid. Thus, DL still improves the performance but not as much as the case of using ReLU activation functions in all four hidden layers.

*4.5. Final Remarks and Future Directions*

In this work, we considered a generalized way of replacing the linear lead-lag-type controller of a second-order system by DL to represent a wide range of control systems. Other linear controllers (such as PID) can be replaced by DL using similar analysis.

All of the control systems and plants considered in this paper are linear systems, in the sense that they all obey the superposition property. Furthermore, uniform sampling that follows Shannon–Nyquist Theory is considered, both in the conventional control sub-systems and in their DL counterparts. As the topic of DL control is developing, we expect future research to handle non-uniform sampling and non-linear DL control subsystems. Please note that nonlinear systems are often approximated by LTI systems using various techniques, e.g., via approximating the nonlinear transfer function near an operating point [28], or via local coordinate transformation [29]. Hence, this work will be the building block of designing DL approximations for nonlinear control systems. On the

other hand, since DL controller exhibits better performance than existing linear systems with the additional merit of robustness against parameter changes, it is expected that DL control will be a better choice for approximating nonlinear systems than existing methods.

Another direction for future investigation would be to consider DL for real-time control (RTC), which is time-constrained control to handle time-varying (or predicted) parameter changes either periodically or via triggered actions to activate a suitable subsystem from a bank of predesigned control subsystems. However, due to the robustness of DL control subsystems against parameter changes, as explained in Section 4.3, DL control is already enabling RT control. If further possible changes are expected to be incorporated into the plant transfer function so that a bank of pre-designed DL control subsystems is required, then DL control would require far fewer triggers (hence, reduced complexity) as compared to the conventional RT control subsystems. On the other hand, as DL control is emerging, it would be too early at this stage to talk about unsupervised training to handle time-varying parameters in DL real-time control (DL-RTC).

Despite the detailed study of the robustness of the proposed DL control subsystem against parameter changes, as explained in Section 4.3, further research is required to develop the full theory of Robust DL Control.

## 5. Conclusions

A deep learning controller can efficiently replace a conventional controller in a feedback control system if trained offline with the input-output signals of the conventional controller. It has been shown that if the DL controller is sufficiently deep, it can outperform the conventional controller in terms of settling time of the step response of the tested second-order system. In addition, no re-training is needed under different reference input magnitude or in case of system parameter change. Under system parameter change, the conventional controller needs to be redesigned for comparable performance with the DL controller. Another performance indicator for feedback control systems, namely the steady state error, was also shown to improve using DL controllers. A call for a future direction of robust DL control is also presented.

**Author Contributions:** Conceptualization, N.A.S.A. and Z.M.H.; methodology, N.A.S.A.; software, N.A.S.A.; validation, N.A.S.A. and Z.M.H.; formal analysis, Z.M.H. and N.A.S.A.; investigation, N.A.S.A. and Z.M.H.; resources, N.A.S.A. and Z.M.H.; writing—original draft preparation, N.A.S.A. and Z.M.H.; writing—review and editing, Z.M.H.; visualization, Z.M.H.; supervision, Z.M.H.; project administration, Z.M.H. All authors have read and agreed to the published version of the manuscript.

**Funding:** This research is partially funded by Edith Cowan University via the ASPIRE Program.

**Data Availability Statement:** The MATLAB code is available from the authors on request.

**Acknowledgments:** The authors would like to thank Edith Cowan University for supporting this project. Thanks to the reviewers for their insightful comments. Sincere thanks to the Academic Editor and MDPI Office for their valuable guidance and immediate attention throughout the review process.

**Conflicts of Interest:** The authors declare no conflict of interest.

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
