# Peer review of "Deep Learning Control for Digital Feedback Systems: Improved Performance with Robustness against Parameter Change"

_electronics, doi:10.3390/electronics10111245_

Round 1
Reviewer 1 Report
The paper presents a pioneering application of deep learning network that can control a digital feedback system. The proposed network can replace classical electronic control sub-system, with less error and overshoot.
However, a major revision is necessary to clarify the following points:
- The authors have chosen rectified linear unit (ReLU) as activation function for the hidden layers, and a linear activation for the output layer.
- There should be a clearer reasoning for using those activation functions. If applicable, the authors should present a numerical comparison with other activation functions.
- The authors should also comment on the possibility of using different activation functions for different hidden layers.
- Robustness against parameter change is an important feature of the proposed work. The authors should present this study under separate sub-titles, i.e. performance under change of plant gain, and performance under change of plant poles.
- While a significant change of gain has been considered, the limits of pole change are still not clear. Another example of a larger pole change would be necessary.
- The authors used a network buffer size of 8 for the case of two hidden layers. However, the number of input nodes for the case of four hidden layers is not clear.
Reviewer 2 Report
In the paper interesting research outcomes of deep learning control for digital feedback systems are presented. The paper is technically sound and describes in a comprehensive way the obtained results. The paper provides sufficient background. The paper is good organized and written on a satisfactory level. However, in my opinion, the Authors’ contribution should be more clearly presented (e.g., in the Introduction section).
Reviewer 3 Report
The paper presents a deep learning based contorller design.
The paper presents a single 'conventional' controller design and compares it with a neural network trained controller, a bakc propagation network is used to train this controller.
The introduction of the paper doesn't contain enough information about the background, the state of the art and the motivation of the paper. It must be show some similar solutions and their advantages/disadvantages from neural network based controller design. The theoretical background of the applied neural network methods are very outdated, there should be 5-10 new citations added to this introducation to be able to show the novelty of the paper. The definition of the robustness should be clarified, it used too generarly in the title, or some robustness measures should be mentioned in the paper (https://doi.org/10.3390/app10196653).
Round 2
Reviewer 1 Report
All questions raised by the reviewer have been fully addressed. I therefore recommend that the revised manuscript be accepted for publication in this journal.Author Response
Thank you for your time and positive opinion. Your insightful comments helped in improving our paper significantly.
Reviewer 3 Report
The authors answered all of my questions.
Author Response
Thank you for your time and insightful comments.